# Research on the Application Status of Machine Vision Technology in Furniture Manufacturing Process

**Rongrong Li** [1,2,*] **, Shuchang Zhao** [2] **and Bokai Yang** [2]

1  Co-Innovation Center of Efficient Processing and Utilization of Forest Resources, Nanjing Forestry University, Nanjing 210037, China
2  College of Home Furnishings and Industrial Design, Nanjing Forestry University, Nanjing 210037, China
*  Correspondence: rongrong.li@njfu.edu.cn

**Abstract:** Machine vision technology was integrated into the manufacturing workshop, to achieve an effective and high-quality production mode for furniture manufacturing. Machine vision can be used for information collecting, quality detecting, positioning, automatic sorting, intelligent monitoring, etc., which largely make up for the shortcomings of poor quality, low precision, low efficiency, and high labor intensity of manual operation. In this study, the method of systematic literature review was applied, and 128 relevant literatures in the field of machine vision application in manufacturing were retrieved and screened from 2011 to 2022. Statistical analysis was carried out on the extracted application directions and related technologies. The current status of machine vision technology's implementation in furniture manufacturing was summarized. In view of the new demand of the rapid development of intelligent manufacturing, the challenges, faced by machine vision, were also summarized. To build a more intelligent, comprehensive, and effective manufacturing workshop for wooden products, cutting-edge technologies, such as deep learning and 3D point cloud, must be further integrated into machine vision. This study can efficiently assist the pertinent practitioners in furniture manufacturing in quickly grasping the pertinent technical principles and future development directions of machine vision, which would be benefit for accomplishing intelligent manufacturing.

**Keywords:** machine vision; furniture manufacturing; quality detecting; automatic sorting; systematic literature review





## 1. Introduction

Machine vision (MV), which utilizing the optical tools and computer technology, could be applied to collect object data for processing and comprehension as well as to simulate human visual function [1]. It has been widely used in automobile, transportation, pharmaceutical, aerospace, military, and many other fields [2]. Nowadays, it has attracted much attention in the field of intelligent manufacturing. MV as an interdisciplinary technique, encourages traditional manufacturing processes to become more intelligent. It requires expertise in a variety of fields, which including optical imaging, signal processing, image processing, machine learning, and deep learning [3]. Academics have researched the digital image processing in considerable depth since the 1950s, when the concept of MV was originally proposed [4]. From the 1970s to the 1990s, a range of disciplinary procedures was combined with MV to enrich and implement its theoretical structure [5,6]. In the 21st century, with the rapid development of artificial intelligence, machine learning and deep learning technology, MV can achieve higher accuracy and more applications. It has been commonly utilized in furniture workshops as the application scope of MV has gradually expanded [7,8]. The application of MV in furniture manufacturing has hastened the transition of modern furniture manufacturing workshops from manual to intelligence manufacture [9]. The advantages of MV include e its high detection accuracy, quick speed,

and strong coordination. It can increase the intelligence of wooden products manufacturing workshops, accelerate production, improve detection accuracy and reduce accident risk [10].

## 2. Methods

The method of system literature review (SLR), which proposed by Tranfield et al., (2003) [11], was applied to retrieve, sort out and analyze the relevant literature, and finally summarized the application status and technical realization methods of MV in the furniture manufacturing process. SLR includes the following steps.

Step 1 is to define the boundaries of the concept. Considering the objectives and research questions of the review, conceptual boundaries are defined [12]. Consistent with our goal of exploring the application of MV in the furniture manufacturing industry, we have defined neither the MV used in this paper nor any restrictions on the furniture manufacturing process that applies these technologies. Our research therefore includes papers on MV applied to other industries, both independently and in combination with other advanced manufacturing technologies.

Step 2 is to determine the retrieval criteria. Search database: the web of science core collection was used. Search time range: 2011–2022. Search keywords: 10 keywords related to MV and 20 keywords related to the furniture manufacturing industry (as shown in Figure 1) were selected for combination, to determine the papers that may be included in the review. MV keywords included the key equipment and key technology. The keywords related to the furniture manufacturing industry were determined according to the actual demand of furniture manufacturing workshops. We searched the keywords related to MV and the furniture manufacturing industry in the title of the literature, and a total of 2601 literatures were retrieved.

Machine vision OR
Computer vision OR
Vision OR Depth learning
OR Machine learning OR
Camera OR Image processing
OR Digitization OR 3D cloud
OR Edge detection

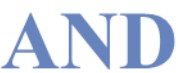

Wood OR Furniture OR Manufacturing OR
Tool monitoring OR Equipment status monitoring OR
Information acquisition OR Quality testing OR Size
OR Hole OR Defect OR Colour OR Location OR
Sorting OR Workshop monitoring OR Material
opening OR Edge banding OR Row drill OR Storage
OR QR code recognition OR Barcode recognition

**Figure 1.** Keywords used for the papers collection in the selected databases.

Step 3 is to select and include relevant literature. By selecting the categories of web of science, 219 literatures related to these categories were further screened. After reading the abstracts of these 219 literatures, some literatures that are not relevant to the content of this study were excluded, such as medical diagnosis, business prediction model and camera calibration, and 128 literatures were retained at last.

Step 4 is to extract, analyze and synthesize key information such as application model and key technologies in the literature. According to the extracted application model, the MV application was summarized into four aspects, such as information collecting, quality detecting, positioning and automatic sorting, and intelligent monitoring. Table 1 showed the overview of the literature and key technologies.

**Table 1.** The overview of the literature and key technologies.

| Application | Details | Image Processing | Machine Learning | Deep Learning | 3D Vision |
|---|---|---|---|---|---|
| Information collecting (*n* * = 2) | • Barcode recognition<br>• Quick Response (QR) Code recognition | 2 | / | / | / |
| Quality detecting (*n* * = 97) | • Dimensional defects (Length, width, thickness, etc.)<br>• Dimensional accuracy of hole processing (Diameter, location, depth, etc.)<br>• Surface defects (Solid wood defects, bumps, scratches, etc.)<br>• Color difference (Color grade classification) | 52 | 27 | 48 | 7 |
| Positioning and automatic sorting (*n* * = 26) | • Positioning<br>• Sorting<br>• Disorderly sorting | 13 | 1 | 8 | 10 |
| Intelligent monitoring (*n* * = 20) | • Status of personnel<br>• Device status<br>• Tool wear condition<br>• Status of inventory | 6 | 5 | 7 | 6 |
| | Total | 73 | 33 | 63 | 23 |

* *n* represents the number of relevant literatures.

Step 5, results and discussion.

## 3. Results and Discussion

Figure 2 summarized four kinds of the application of MV in furniture manufacturing industry. According to the findings of the collation and analysis, the application aspects with little literatures should be searched separately, and the time and scope restrictions should be relaxed. The practical application technology of these four aspects was described in details as follows.

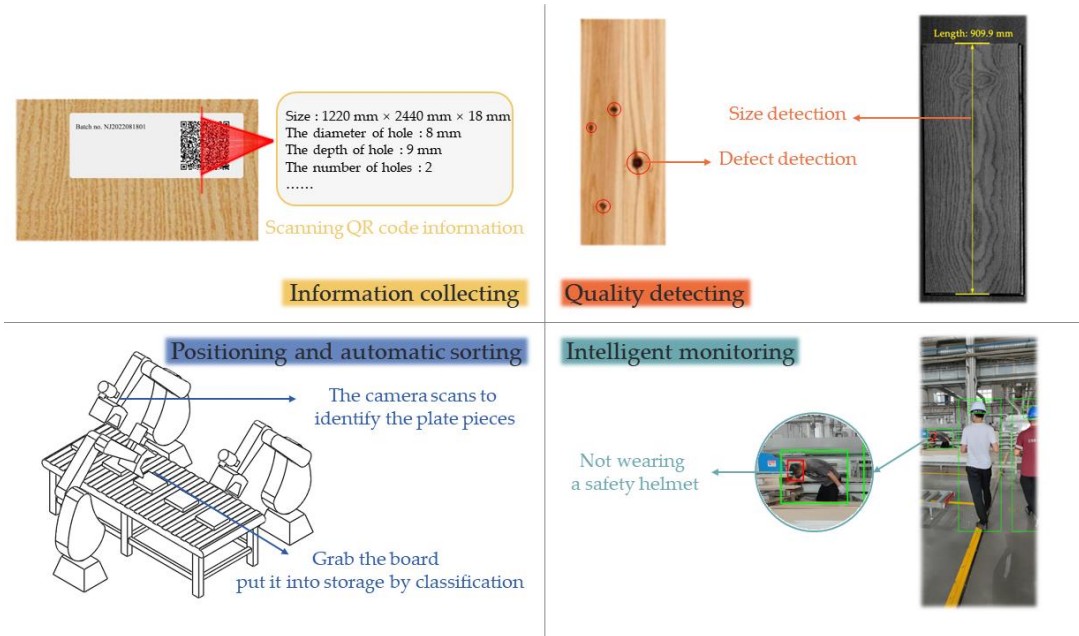

**Figure 2.** Applications of MV in furniture manufacturing workshop.

### 3.1. Information Collecting

The initial stage of furniture production processing is information gathering. As shown in Figure 3, MV can immediately locate the coding position in coding information collection line during the furniture manufacturing. It has the advantages of shorter positioning time, smaller difficulty of coding recognition, higher accuracy, which adapt to real-time online detection. MV was employed by Lin et al., (2011) and Xu et al., (2018) to batch recognize encoded information in high-speed motion, and both the detection accuracy and efficiency were higher than manual recognition [13,14]. Barcodes with slight defects can also be recognized by MV, and the recognition accuracy could reach to about 100% [15]. The encoded information on the furniture components and parts, such as present position, time, and forthcoming operation data, could be captured by MV and uploaded to guide the master database for workpiece's further processing or troubleshooting. The gathered data can also be used to analyze the inventory status and compile statistics, enabling automatic storage information management, and assist to control the production process [14].

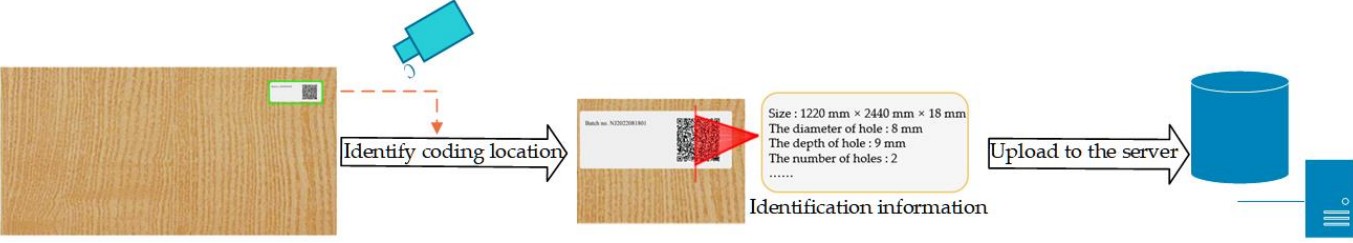

**Figure 3.** Flow chart of information collection.

### 3.2. Quality Detecting

The quality detecting during furniture manufacturing process is to ensure the quality of furniture components and parts and prevent the outflow of defective products. For the panel furniture, the objects that need to be examined include dimensional defect, dimensional accuracy of hole processing, surface defect, and color difference. At present, manual detection is still the main method in furniture workshops. Color and surface defects are identified by employees' experience, and the dimension and position of the hole are mostly measured by tape [16]. The detection efficiency is low, the error rate of the check process is high, and there is also a risk of scratching the panel during manual detection [17]. The application of MV inspection can make up for the deficiencies of manual inspection, achieving higher efficiency and lower error rate, and making the measurement data more precise during the non-contact inspection process.

In the beginning of quality detecting technology based on MV, three steps of image acquisition, image processing, and image segmentation are needed to make preliminary preparation [18].

(1) Image acquisition

Image acquisition is an important link in the realization of MV quality inspection technology. Acquiring high-quality images can reduce the difficulty of subsequent image processing. In the field of traditional machine vision, two visible light sensors, either charge-coupled devices (CCD) and complementary metal-oxide-semiconductor sensors (CMOS [19]), are often used. Their photoelectric conversion principle is the same. Compared to CCD, CMOS has the advantages of low power consumption, but it pro-duces more noise. CCD sensors have high sensitivity and low read noise, but consume high power. Infrared photoelectric sensors can capture thermal data and play an important role in the measurement of temperature. Compared with optical sensors, it is more suitable for all-weather work, and it is more effective in detecting long-distance targets. In recent years, sensors that incorporate 2D materials (metals, semi-metals, semiconductors and insulators, etc.) have become strong competitors to traditional sensors due to their thinness

and wide detection band. In the future, they may become the main sensors for the next generation of machine vision [20].

(2)    Image preprocessing

Image preprocessing includes image denoising, image enhancement and other operations. Common denoising methods include median filtering, mean filtering, Gaussian filtering, bilateral filtering, etc., which can effectively reduce image noise [21]. The commonly used image enhancement methods are gray transformation, color space change, etc., which can effectively reduce the interference of unimportant information, such as image background on subsequent image processing. In recent years, deep learning has been applied to image noise reduction techniques and performed well in practical applications [22]. Chen et al., Zhang et al., and Cruz et al., apply convolutional neural network (CNN) to denoise images, and the effect is good in practical denoising applications, which is better than traditional methods [23–25].

(3)    Image segmentation

Image segmentation is an important basis for feature extraction. In the actual application process, according to different usage scenarios, selecting an appropriate image segmentation algorithm can effectively improve the quality of the segmented image and be better applied to subsequent operations. Traditional image segmentation methods mainly include threshold-based segmentation methods, edge-based segmentation methods, region-based segmentation methods, and specific theory-based segmentation methods [26]. In recent years, image segmentation based on deep learning has become the mainstream of research. Shervin et al., summarized the application of deep learning in image segmentation in detail, introduced image segmentation models based on neural network, Encoder-Decoder, Multiscale and Pyramid Network, Dilated Convolutional, etc., and evaluated the segmentation performance [27].

Then, different algorithms are selected for practical applications. At present, the quality detection of furniture parts mainly includes dimensional defect detection, hole processing quality detection, surface defect identification and color difference detection, as shown in Figure 4. Dimensional defect detection is used to determine the length, width, and thickness [28]. Hole processing quality detection can identify the hole's location, number, dimension, and other characteristics on the products components and parts [29]. MV can also automatically identify and classify surface defects in wooden products, such as scratches, potholes, dead knots, slip knots, and wormhole, etc. Color is one of the key parameters, which affects the products appearance quality, so large color difference is not allowed. MV could be applied for color sorting, and the same color level of product will be put into the same category [30]. Figure 5 summarized the quality detecting technology implementation process.

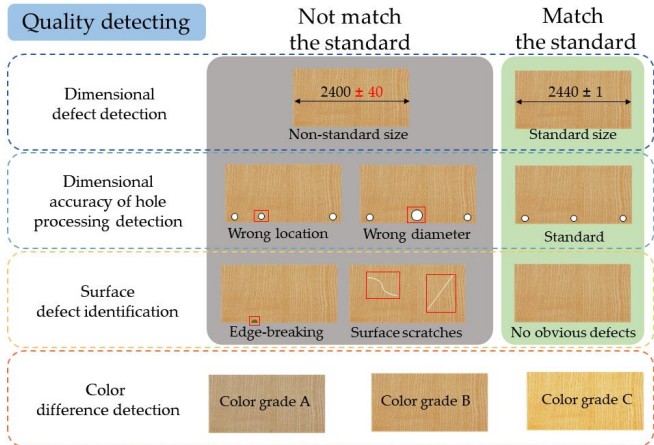

**Figure 4.** Application of quality detecting.

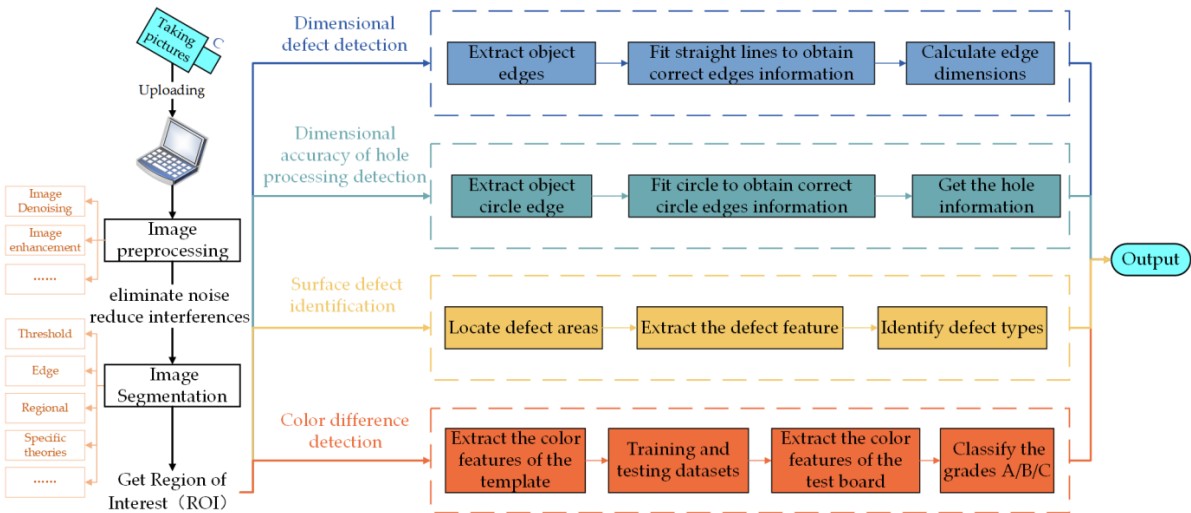

**Figure 5.** Technical roadmap of quality detecting technology based on MV.

### 3.2.1. Dimensional Defect Detection

The dimensional defect detection technology based on MV can accomplish the detection work with high real-time, high accuracy, high efficiency, low error, and non-contact. The most common dimensional defect detection method involves pre-processing the image, using the Hough transform (HT. Its basic idea is to convert the detection problem from the image space into the parameter space, complete inspection tasks using straightforward statistics accumulated in the parameter space, and describe the edge of the image by some form of parameter that satisfies most of the edge points [31].) to identify the workpiece's outline information, finding the edge line and transforming it, rotating the image of workpiece's edge to the horizontal position in accordance with the slope of the line, and then calculating the actual physical distance between the pixels using camera calibration data. The measurement data is uploaded to the database, and compared with the database data. If the error is within the range, the workpiece will be retained and entered the next process. Otherwise, it will be eliminated. However, the Hough transform method has a large amount of calculation and is seriously affected by noise [32], which requires high-quality collected pictures. Only applying the Hough change cannot meet the variety of detection needs due to the complexity of workpiece and the enhancement of dimensional defect detection requirements. In recent years, many scholars have made more and more in-depth research on dimensional defect detection by MV. Edge detection algorithms (such as the Canny operator, and Sobel operator) and particle swarm algorithms have been proposed and applied to the field of dimensional defect detection [33,34]. The detection range extends from length and width to angle, radian, diagonal length, and even thickness. Table 2 summarized and compared the advantages and disadvantages of several commonly used edge detection algorithms.

**Table 2.** Comparison of edge detection algorithms.

| Algorithm | Advantages | Disadvantages |
|---|---|---|
| Canny | <ul><li>High precision</li><li>Accurate</li><li>Anti-noise</li><li>Suppresses false edges</li></ul> | <ul><li>Large amount of calculation</li><li>Poor real-time performance</li><li>Less edge detail</li><li>Poor adaptability</li></ul> |

**Table 2.** *Cont.*

| Algorithm | Advantages | Disadvantages |
|-----------|-----------|---------------|
| Sobel | • Simple<br>• Directional<br>• Anti-noise | • Poor accuracy<br>• Easy to miss edge<br>• Wide detection edge |
| Roberts | • Simple<br>• Fast<br>• High precision | • Easy to miss edge<br>• Wide detection edge<br>• Many limitations |
| Prewitt | • Anti-noise<br>• Fast | • Poor accuracy<br>• Blurred edges |

3.2.2. Dimensional Accuracy of Hole Processing Detection

Hole information, which detected by MV, can greatly reduce the time requirement, improve detection accuracy, and optimize the detection process. The commonly used detection method of hole characteristic parameters is similar to the dimensional defect detection. Firstly, the region of interest (ROI) will be extracted. Then, using HT [35] to detect the edge information of the hole on the ROI area. At last, the ellipse fitting will be used. The fitted circle presents the hole in products parts. The hole information will be compared with the database data, and the reasonable error is considered qualified. Although, the detection method based on Hough change is traditional, it still has great application value. Scholars improved the basis of HT and proposed probabilistic HT [36], randomized HT [37], fuzzy HT [38], Curvature aided HT [39], etc., which can be better applied to circle detection. Li et al., used a semi-supervised learning algorithm for circular hole detection [31]. This method can meet the circular hole detection requirements of plates with complex textures. The measurement accuracy can reach to 0.03 mm. However, the existing literature mainly focused on the detection of hole diameter information, and seldom concerned the depth information of hole. 3D point cloud technology solves the problem of hole depth measurement from the 3D level. The edge feature points of the hole are retrieved and refined based on the 3D point cloud data of the hole [40]. To acquire data on hole distance and aperture, the feature points are parameterized by a space circle in the final step.

3.2.3. Surface Defect Identification

Surface defect identification by MV includes three steps of image pre-processing, defect area localization, and defect feature extraction [41]. Firstly, image pre-processing includes eliminating noise and other interference information, enhancing image contrast, and making the features in the image salient. Secondly, different methods of the model comparison method and image segmentation method, will be used to locate the defect areas for the different objects. Thirdly, some relevant algorithms would be selected to analyze the location area, extract defect features, and identify and classify defect types. At present, the surface defect identification methods based on MV are mainly divided into three categories: defect detection based on image processing, defect detection based on machine learning, defect detection based on deep learning.

Surface defect detection, which based on image processing, applies an image segmentation algorithm to extract defect surface features. Canny and Sobel operators are the more commonly used algorithms to extract defect edges. And the feature extraction or template matching is applied to identify defects [42]. This defect recognition method is simple, widely used, and has good robustness. But it is difficult to deal with some complex changes in the defect and has no autonomous learning ability. It is only suitable for a single fixed detection target. In recent years, the convolution neural network (CNN), deep belief network (DBN), and residual neural network model (ResNet) that have deep learning capabilities are gradually applied for defect detection [43–45]. Compared with image processing and machine learning, these deep learning methods have higher detection efficiency and the ability to learn. However, their calculation is complicated, many datasets

are needed to be trained [46–48]. Kamal et al., (2017) utilized gray level co-occurrence matrix and laws texture energy measures as texture feature extractors, and chose feed-forward back-propagation neural network as classifier to identify wood defects. The results showed that that method had an accuracy of 84.3% [49]. In Table 3, the commonly used defect recognition algorithms in recent years were summarized and the advantages and disadvantages of these different algorithms were analyzed.

**Table 3.** Comparative analysis results of some commonly used defect recognition algorithms based on MV.

| Detection Method | Algorithm Model | Advantages | Disadvantages |
|---|---|---|---|
| Image processing | • Feature extraction<br>• Template matching | • Simple<br>• Wide application range<br>• Better robustness | • Difficult to handle changes<br>• No self-learning ability |
| Machine learning | • Support Vector Machine<br>• Decision Tree | • Strong adaptability<br>• Strong universality<br>• Good robustness | • Low generalization<br>• Susceptible to noisy data |
| Deep learning | • Convolutional Neural Networks (CNN)<br>• Deep Belief Networks (DBN)<br>• Fully Convolutional Networks (FCN)<br>• Autoencoder | • Wide application range<br>• Strong expansibility<br>• High accuracy rate<br>• Automatically extract implicit features | • High data volume requirements<br>• Complicated calculation<br>• High hardware requirements |

### 3.2.4. Color Difference Detection

Based on MV, fast, efficient, and low-error color difference detection can be realized [50–52]. The color difference detection based on MV is different from the above three aspects. It needs to use a color camera to collect color images. Commonly used detection methods need to pre-select multiple groups of boards with different color grades, extract the values of the color space (The color space models mainly include HSV, RGB, HSI, CHL, LAB, CMY, etc.), and perform training, testing, and verification. At the beginning of the application, image acquisition and pre-processing are executed. Then, locate the workpiece area, remove the interference of the background color to the detection, and extract the color features. At last, the classifier algorithm will be applied for color classification. Color difference detection involves deep learning algorithms and requires many data sets for training, testing, and verification. The more sufficient the data set, the more significant the grading effect will be. The color information the wood board, which captured by color camera, will be affected by the ambient light condition. In the factory actual application, the change of ambient light greatly interferes with the results of the color difference inspection for furniture. Placing a hood around the camera and lights can effectively reduce the impact of ambient light condition on the color information collection. In the actual application process, the combination of algorithms such as extreme learning machine (ELM), multi-layer perceptron (MLP), and gray wolf algorithm is more competent to achieve the optimal recognition effect [53–55].

### 3.3. Positioning and Automatic Sorting

The MV positioning technology can realize the accurate determination of the work-piece position [56], which is convenient to analyze and use the workpiece position information for subsequent processing operations. For example, automatic sorting of furniture parts, precise spraying of paint spray gun, etc. are all need MV technology to assist in their operations. After the quality detecting process, the products will be sorted and put into storage. The manual sorting process requires workers to move the workpiece to the specified position after validating the information obtained by scanning the board's bar code or electronic label. A 250-person furniture manufacturing workshop can produce about 20,000 parts per shift (8 h). If there are 10 sorting employees, each sorting employee needs to sort an average of 2000 boards per shift. It is labor-intensive, inefficient, and costly, and cannot accommodate the current trend of mass customization. MV can calculate the three-dimensional coordinates of objects in the real world, and guide the manipulator to

locate the plate [57]. MV can significantly improve sorting efficiency, reduce the error rate, and reduce labor costs effectively [58]. Wu et al., combined robot technology with MV to identify and sort the sorting objects at a high speed, which improved the automation and intelligence level of flexible manufacturing production lines [59–61].

### 3.4. Intelligent Monitoring

The application of MV in monitoring workshops is to use computer equipment and learning algorithms to quickly analyze the specific content information in the video [62]. As shown in Figure 6, it is primarily used to monitor three aspects, including employee behavior, storage status, and device status.

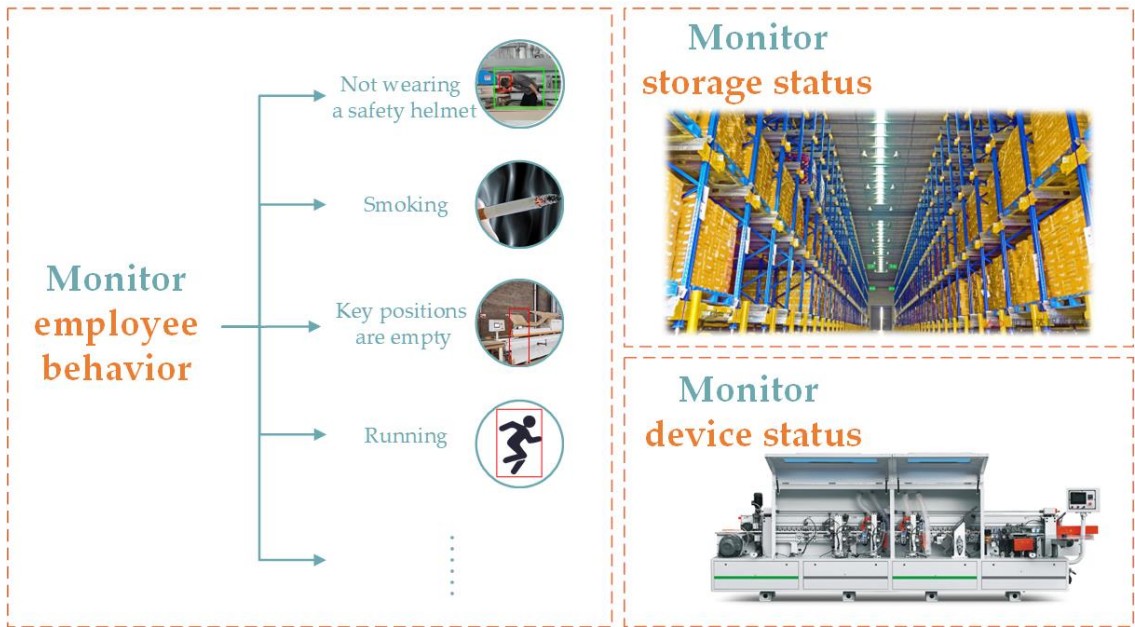

**Figure 6.** Application of intelligent monitoring.

Furniture manufacturing workshops include workers and equipment. Workers have a certain risk of injury during some operations, and if these operators do not follow conventional operating procedures, a serious mishap would be quickly occur. Nowadays, some smart monitoring devices are often used to monitor and evaluate employee behavior, such as monitoring whether the employees wear safety helmets and whether the employees have dangerous behaviors. The data of workers' long-term actions are gathered and examined by MV, and these data would be used as a reference for guiding the workers' behavior. Currently, intelligent monitoring technology mostly uses the YOLOv4 algorithm to locate and evaluate targets and to send out early warning signals whenever a situation deviates from previously established standards (such as workers are not wearing safety helmets, smoking, not in the key positions) [63,64]. MV also could be applied for storage status monitoring in the workshop. The monitoring of the inventory can reflect the inventory status in real-time, and alarm in the case of inventory backlog or shortage, so that the staff can timely supply the warehouse [65]. The state of equipment affects the normal operation of production, which could be monitored by visual monitoring technology, such as monitoring equipment tool wear [66–69], equipment conveyor belts [70]. Real-time understanding of device status could effectively reduce the downtime of machine failure.

*3.5. Limitations of MV and Directions for Future Work*

3.5.1. Limitations of MV

(1)  Limited by the site environment

Most of the development and research of MV system is completed in a specific environment, which has poor adaptability and versatility. Once the environment changes (such as light source, conveyor belt color, etc.), the speed and accuracy of the detection algorithm will be affected. In the future, MV technology should be improved by hardware and software optimization, to achieve stronger adaptability and versatility [71].

(2)  Limited by device performance

MV is also limited by equipment performance, which includes two aspects of camera performance and computer computing power. The limited observation range of the camera, the distortion produced by the lens, the quality of the sensor, etc., affect the quality collected image. The performance of the computer graphics card has a great influence on the speed of image processing. Insufficient performance of the computer graphics card will affect the running speed of the image processing algorithm, resulting in a short-term stagnation of the entire workshop production line, making it difficult to improve industrial production efficiency [2].

(3)  Limited by the diversity of detection targets

The needs in the production process of the workshop are diverse, and the single system is difficult to meet multiple detection requirements [72]. Therefore, most of the relevant research focuses on a single detection requirement. Under the same detection requirement, there are various objects to be detected. In surface defects identification process, there are various types of defects, which is difficult for MV systems to extract all defect features. For the ambiguous defect characteristics, the processing capability of the MV system is still insufficient.

(4)  Limited by the scarcity of talents

Most workers in the manufacturing workshop do not understand the operating principle of the MV system and have no ability to independently deal with equipment failures. They can not adjust the relevant parameters according to the needs of the workshop. With the popularization of education and the development of the MV industry, the number of MV talents is gradually increasing, and the impact of this problem is gradually weakening.

3.5.2. Directions for Future Work

(1)  3D MV technology

With the increase of inspection requirements, 2D MV has been difficult to meet the needs of workshops. Structured light cameras [73], binocular and multi-eye vision [74], and other technologies that can obtain 3D information of objects are gradually applied in factories. 3D machine vision can break the bottleneck of the detection range of 2D machine vision, and meet the needs of industrial applications. It has gradually become the future development trend of the MV. In the future, using 3D MV combined with robots, AI and other new technologies can create an intelligent workshop.

(2)  Combination of traditional MV and deep learning

Traditional MV often requires the assistance of machine learning (Support Vector Machine, Decision Tree, etc.) and basic image processing algorithms (OSTU, Hough transform, Canny, etc.) [75]. Sometimes, it is not competent for changing detection objects. The MV based on deep learning is more adaptable to change and provide greater versatility in practical applications, but it requires a large amount of labeled data for training. The combination of traditional methods and deep learning can make up for each other's shortcomings, reduce the difficulty of algorithm processing, and improve the running speed and performance [1].

## 4. Conclusions

MV is applied to the monitoring of the workshop to ensure personnel safety and equipment efficiency, information gathering at each manufacturing link to ensure manufacturing process transparency, check product quality to guarantee that no defective items are released, and upload alarm information for repair when the problems are discovered to prevent recurrence. Automatic sorting of furniture parts could accomplish products classification and storage. The application of MV significantly increases the production management condition in furniture manufacturing workshops, as well as the degree of automation and production efficiency. In the future, 3D MV could combine with other advanced manufacturing technologies to create a safe, intelligent, and efficient manufacturing workshop.

**Author Contributions:** Conceptualization, R.L.; methodology, S.Z.; investigation, S.Z. and B.Y.; resources, S.Z.; data curation, S.Z.; writing—original draft preparation, R.L. and S.Z.; writing—review and editing, R.L.; visualization, B.Y.; funding acquisition, R.L. All authors have read and agreed to the published version of the manuscript.

**Funding:** This research was funded by the National Natural Science Foundation of China (32201642), the project from International Cooperation Joint Laboratory for Production, Education, Research and Application of Ecological Health Care on Home Furnishing.

**Institutional Review Board Statement:** Not applicable.

**Informed Consent Statement:** Not applicable.

**Data Availability Statement:** Not applicable.

**Conflicts of Interest:** On behalf of all authors, the corresponding author states that there is no conflict of interest.

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
