# Peer review of "Research on the Application Status of Machine Vision Technology in Furniture Manufacturing Process"

_applsci, doi:10.3390/app13042434_

Round 1

Reviewer 1 Report

This manuscript focused on the researching of the application status of machine vision in furniture manufacturing process control. The application types of information collecting, quality detecting, positioning, automatic sorting, intelligent monitoring was summarized. The structure and text writing are reasonable. It was suggested to publish after minor revision. My comments were shown as following.

(1) The part of “Abstract” need to be improved.

(2) “In the 21st century” should be revised to “In the 21st century”. And the text should be checked again.

(3) “The system literature review (SLR) method” is suggested to revised to “The method of system literature review (SLR)”.

(4) In table 1, QR code, the abbreviations need to be explained when they first appear.

(5) In table 1, Size, Hole, Defects, Color should be explained.

(6) Figure 4 should be revised.

(7) “Defect identification by MV includes three steps:” is suggested to revised to “Defect identification by MV includes three steps of”.

Reviewer 2 Report

The manuscript titled “Research on the Application Status of Machine Vision in Furniture Manufacturing Process Control” focused on the current status of machine vision technology's implementation in furniture manufacturing. This manuscript is well structured and well written, which is easy to follow. The figures and tables are neat and easy to understand.

Generally, I suggested this manuscript can be accepted for publication after minor revision.

(1) At the end of abstract, it should be included one or two lines that how it is going to benefit the scientific community or what are the readership of this paper.

(2) Keywords. I think “surface detecting” is more reasonable than “quality detecting”.

(3) Provide more information for the “Hole Detection”.

(4) It is hard to follow the “ the quality detecting technology implementation process” in Fig. 5, make more explanation.

(5) It is suggested to supplement the commonly used algorithm types in quality inspection.

(6) Is manual detection  still the main method for furniture manufacturing? What is the difficulty of machine vision promotion?

(7) Reference and explanation for Hough change.

(8) et al. shoud be in italics, et al.

(9) It is hard to follow the “some intelligent monitoring equipment are frequently used to monitor employee behavior, assess worker behavior, determine whether their attire complies with regulations, and evaluate whether the operation is standard”, please modify it.

(10)“There are still some limitations to the application of MV in furniture manufacturing” should be “There are still some limitations for the application of MV in furniture manufacturing.”

(11) The format of references needs to be modified. Some pages are missing.

Reviewer 3 Report

This research summarized and reviewed the application status of machine vision in furniture manufacturing process. As we known, MV is an advanced technology for information collecting and processing, which widely applied in intelligent manufacture. As a traditional manufacturing industry, furniture industry is facing the opportunity and adjustment of transformation and upgrading. This review paper will be benefit for the researchers and industry practitioners to better understanding of MV technology. I have some suggestions to the authors to improve this work.

1. The title is suggested to revise to “Research on the application status of machine vision technology in furniture manufacturing process”.

2. Line 11, please delete “workshops”.

3. The structure of abstract is suggested to improve. The research method and results need to be added.

4. Line 80 is suggested to list below line 77.

5. Line 82, “the four kinds” is suggested to delete “the”.

6. Figure 4 should be revised. What is the meaning of unqualified and qualified?

7. Figure 5, the text needs to be revised.

8. Table 2 and Table 3, the period is not needed.

9. The case should be uniform. Such as 3.2.1. Size Detection, 3.2.3. Defect identification, etc.

10. The title of figures do not need period.
